# An Approach to the Automatic Construction of a Road Accident Scheme Using UAV and Deep Learning Methods

**DOI:** 10.3390/s22134728

**Published:** 2022-06-23

**Authors:** Anton Saveliev, Valeriia Lebedeva, Igor Lebedev, Mikhail Uzdiaev

**Affiliations:** St. Petersburg Federal Research Center of the Russian Academy of Sciences (SPC RAS), St. Petersburg Institute for Informatics and Automation of the Russian Academy of Sciences, 39, 14th Line, 199178 St. Petersburg, Russia; saveliev.ais@yandex.ru (A.S.); lebedev.i@iias.spb.su (I.L.); uzdyaev.m@iias.spb.su (M.U.)

**Keywords:** road accident, UAV, deep learning model, segmentation, measure the distances, reconstruction of a road accident

## Abstract

Recreating a road traffic accident scheme is a task of current importance. There are several main problems when drawing up a plan of accident: a long-term collection of all information about an accident, inaccuracies, and errors during manual data fixation. All these disadvantages affect further decision-making during a detailed analysis of an accident. The purpose of this work is to automate the entire process of operational reconstruction of an accident site to ensure high accuracy of measuring the distances of the relative location of objects on the sites. First the operator marks the area of a road accident and the UAV scans and collects data on this area. We constructed a three-dimensional scene of an accident. Then, on the three-dimensional scene, objects of interest are segmented using a deep learning model SWideRNet with Axial Attention. Based on the marked-up data and image Transformation method, a two-dimensional road accident scheme is constructed. The scheme contains the relative location of segmented objects between which the distance is calculated. We used the Intersection over Union (IoU) metric to assess the accuracy of the segmentation of the reconstructed objects. We used the Mean Absolute Error to evaluate the accuracy of automatic distance measurement. The obtained distance error values are small (0.142 ± 0.023 m), with relatively high results for the reconstructed objects’ segmentation (IoU = 0.771 in average). Therefore, it makes it possible to judge the effectiveness of the proposed approach.

## 1. Introduction

Despite the many programs and technologies already implemented to collect data on the location of road traffic accidents, the number of accidents recorded worldwide is still growing and, accordingly, the loan on services to provide analysis of the causes of accidents is growing [1,2,3,4,5]. Every year, countless campaigns are conducted around the world that are aimed at raising awareness of drivers about the factors of accidents and attempts to reduce risky behavior on the roads [4,6]. However, accidents still happen. Therefore, it is essential to reconstruct an accident site with maximum accuracy and objectivity to determine the causes underlying the event [7]. The investigation of the scene of an accident, the work of insurance companies, legal aspects, and property damage are stages that are directly dependent on the information collected at the scene. The information obtained is used to determine the causes of an accident, which can potentially have a valuable contribution to the world’s accident statistics and to the subsequent reduction in the number of road accidents. Accurate reconstruction of an accident site will help to identify design flaws of urban and transport infrastructure and/or the behavior of the driver and pedestrian.

Data collection at an accident site is of high importance [8]. The police are not always the first to arrive at the scene of a traffic accident. Typically, their mission is to first ensure that the victims (if any) receive proper medical care, and then to secure the scene of an accident by changing vehicle traffic in the vicinity of an accident while preserving property and evidence. After that, the procedure for registering the accident begins with the identification of evidence; fixing the scheme of the accident, namely, displaying the position and trajectory of vehicles, visible road signs; taking into account conditions at the accident site, such as visibility. In addition to the primary goal of saving lives, restoring normal traffic flow on the roads after the accident is another important task for avoiding even greater social and economic consequences.

Thus, the collection of data at the scene of the accident should be a simple, accurate, and fast procedure. However, data collection and roadmapping is currently mostly a manual procedure performed by people (police) around the world. The accuracy of the map of the scene of a traffic accident depends mainly on the human factor. Difficulties in creating an accident map can be overcome by using small unmanned aircraft (UAVs) capable of autonomous or manual control and deep learning methods to recognize key features of accident [9,10]. UAVs provide data acquisition for building an accurate reconstruction of traffic accidents. Deep learning methods are necessary for segmentation of objects such as vehicles, road markings, signs, and pedestrians. All these data are combined into a single scheme of an accident, according to which it is possible to determine the distance between the objects of an accident and draw a further conclusion about the involvement or non-involvement of drivers in the current situation.

The use of UAV is necessary in cases when it is not possible to promptly collect sufficient and reliable data quickly. If drivers take pictures of an accident, this data will not be enough to establish the cause of an accident and search for the involvement of drivers. The autonomous UAV is able to fully capture the entire scene of an accident, and methods of recognition and calculation of distances between key objects guarantee the analysis of the data obtained with high accuracy. It is assumed that the user, after they have made sure that none of the participants in an accident needs medical assistance, reports an incident to the police and the police sends an equipped UAV to the scene from the nearest station/office. The UAV autonomously collects data and returns to base. If an accident is not serious, i.e., no one was injured, drivers can leave, and the police officers can begin processing the application and the data received. After that, the final decision on the cause of an accident to all participants and their insurance companies.

The approach proposed in this article to the automatic construction of a road accident plan using UAV and deep learning methods includes two aspects: remote collection of data about an accident without the participation of police officers and data analysis [11]. The combination of these technologies makes it possible to analyze a greater number of accidents in the shortest possible time compared to the traditional approach of collecting and building an accident plan when the presence of the police is necessary. Moreover, this combination guarantees high accuracy of construction and marking of an accident plan, compared with the collection of accident data or the sketching of an accident scheme in the European protocol by the accident participants.

The authors in [12] raised questions about the reconstruction of the scene of a traffic accident. They reported about the method of collecting data (aerial photographs) using UAV, as well as comparing the use of UAV of various classes in different weather conditions, lighting, etc. At the same time, all measurements between objects of an accident and the construction of a plan are performed manually. This work shows the advantage of the method used in comparison with the construction of an accident scene with the direct participation of traffic police officers. However, the article does not consider the issues of automatic detection of accident objects, automatic measurement of distances between them, and automatic construction of a complete scene.

The authors of the work [13] explained how to create three-dimensional models of a real road accident site using UAV and photogrammetric procedures. It is on this methodological basis that it is possible to develop a prototype of a road accident investigation system. Since the use of UAV significantly reduces the need for the physical presence of a responsible person, the risk of injury to police officers and other services during the investigation is also reduced. Due to the digitalization of the process, the time required for the reconstruction of a traffic accident diminishes greatly. However, this work also does not address the issues of automatic detection of accident objects on the constructed models. In addition, all measurements of distances between objects on the constructed 3D models are performed manually.

As a tool for analyzing the scene of an accident [14], machine learning methods can be used that solve the problem of segmentation of objects in digital images as well as calculate the distance between key objects. In order to automate the process of analyzing the collected data, in our article we propose to use a neural network that allows one to analyze an accident scene and segment key objects on it. In addition, the article proposes a method for calculating the distances between key objects, which estimates the distances between objects with high accuracy.

Thus, based on the analysis of modern scientific literature, it can be concluded that the issues of automatic detection of accident objects on a 3D scene, automatic measurement of distances between objects, and generation of a complete accident scene have not received enough attention.

The task of detecting objects on the scene is reduced to the task of segmentation (selection of areas belonging to the same object). In modern scientific literature, the issues of segmentation of visual objects in digital images are widely covered, in particular, image segmentation in the tasks of monitoring road traffic and in the tasks of driving cars with an autopilot (self-driving car) [15]. In these tasks, there is a need to perform segmentation of the following classes: roadbed, road markings, traffic lights, road signs, sidewalks, pedestrians, vehicles. The task of segmentation is divided into the following parts: semantic segmentation, instance segmentation, panoptic segmentation.

The result of solving the semantic segmentation problem is the ratio of each pixel in the image to a particular class. As a result, special masks are formed that repeat the shape of an object of a certain class. Morevoer, if two different objects overlap each other, semantic segmentation forms one mask for these two objects.

The task of semantic segmentation of road objects is covered in a number of works. The authors [16] proposed a plain convolutional encoder–decoder architecture, SegNet, to perform semantic segmentation task. SegNet showed 60.1% IoU on the CamVid dataset [17]. In [18], the authors present a convolutional neural network architecture for semantic segmentation of images U-Net, which is an encoder–decoder architecture with skip connections between encoder and decoder parts. This architecture showed an IoU equal to 59,1% on the Cityscapes dataset [19]. Paszke et al. [20] modified the U-Net architecture, presenting a new ENet architecture based on it. The main feature of this architecture is the use of special bottleneck blocks, which can significantly reduce the number of floating-point operations, which ensures real-time operation. On the Cityscapes dataset, this neural network showed IoU, which is equal to 58.3%. In [21], a deep reconciliation neural network of Fully Convolutional Network (FCN) architecture is presented. This neural network has the features of fusion of various scale feature maps by upsampling and convolution operations. When segmenting road objects from the Cityscapes dataset, IoU results were obtained equal to 74.6% [22]. Chen et al. [23] presented a model of semantic segmentation Deeplab, the main feature of which is the use of atrous or dilated convolutions, which allows processing using small-sized convolutional kernels large receptive fields. Another important feature of this model is the use of conditional random fields to build a segmentation map. This model achieves 70.4% IoU on the Cityscapes dataset.

The idea of the instance segmentation task is to simultaneously detect an object in a special localization frame and determine an object mask that identifies its shape. The solution of this problem allows one to correctly determine separate masks for various overlapping objects. Instance segmentation allows one to segment relatively small objects such as cars, pedestrians, road signs; however, it does not allow one to define masks for large background objects such as roadbeds, sidewalks, buildings, etc.

The solution to the problem of instance segmentation of road objects is presented in the work of He et al. [24], where they developed the Mask-RCNN neural network architecture, which is capable of simultaneously performing object detection and segmentation. This architecture is based on the principles of region proposal networks. On the Cityscapes dataset, Mask-RCNN showed 40.3% and 72.4% for AP and AP50, respectively [25]. In [26], the authors present an original neural network architecture for solving the instance segmentation problem. The main feature of the proposed architecture is the use of a recurrent neural network and the attention mechanism [27], a step-by-step process that features maps and performs segmentation of individual objects. On the Cityscapes dataset, this architecture achieved 9.5% AP and 18.9% AP50. The authors of [28] consider the issue of using a discriminative loss function, which is based on the principle of metric learning, for training a neural network of the ResNet architecture [29]. The applied approach made it possible to achieve 17.5% AP and 35.9% AP50 on the Cityscapes dataset. Bai and Urtasun [30] proposed a new neural network architecture, the main feature of which is the watershed transform. The essence of the watershed transform is to calculate the energies of the feature map gradients. This architecture achieves 19.4% AP and 35.3% AP50 on the Cityscapes dataset. Liu et al. [31] suggested using path aggregation network for instance segmentation. This architecture is based on the Feature Pyramid Network (FPN) [32] and differs in the processing of feature maps of different scales using adaptive feature pooling. This architecture makes it possible to achieve AP and AP50 metrics equal to 36.4% 63.1%, respectively, on the Cityscapes dataset. Liang et al. [33] developed a neural network architecture called Polytransform that uses the attention mechanism, segmentation mask polygon embedding, and so-called deforming neural network to fulfill instance segmentation. This architecture approaches 40.1% for AP and 65.9% for AP50.

The task of panoptic segmentation [34] combines the tasks of instance and semantic segmentation: simultaneous segmentation of small objects with separation of overlapping objects (a special term in panoptic segmentation is things) is performed, as well as segmentation of large background objects such as buildings, a road, a sidewalk, etc. (a special term in panoptic segmentation is stuff).

The task of panoptic segmentation of road objects and its solution are described in detail in Kirillov et al. [34], where the authors propose a solution to the panoptic segmentation problem by using the basic architectural principles of FPN to solve both problems: segmentation of things and stuff. The proposed approach achieves 17.6% accuracy according to the Panoptic Quality (PQ) metric on the Mapillary Vistas dataset [35]. The authors of [36] propose a neural network architecture unified panoptic segmentation network (UPSNet) to perform panoptic segmentation. The main feature of this neural network is the fusion of the processing results of things and stuff by a special neural network block named panoptic segmentation head. The result of the model on the Cityscapes dataset according to the PQ metric is 47.1%. In [37], a neural network architecture is proposed, which basically has a synthesis of the FPN architecture and the attention mechanism. On the Cityscapes dataset, this architecture showed a result of 59% according to the PQ metric. Cheng et al. [38] present the Panoptic-DeepLab method. This architecture consists of two parallel branches. The first branch performs semantic multiscale context selection and segmentation of background objects. The second branch performs instance multiscale context extraction and performs instance segmentation by instance center prediction and instance center regression. The authors note the high segmentation results in the Cityscapes dataset, which amounted to 65.5% PQ. In addition, the method showed a high speed of processing video frames (15 fps). In [39], the authors propose an approach to panoptic segmentation that consists of improved recognition of overlapping (occlusion) objects. The essence of the approach lies in the use of special neural network blocks that perform purposeful processing of overlapping objects. The approach showed a PQ metric value of 60.2% on the Cityscapes dataset. Li et al. [40] suggest using a modification of the FCN model to solve the panoptic segmentation problem. The modification consists in using additional neural network blocks called kernel generator, kernel fusion, and feature encoder. These three blocks provide simultaneous segmentation of things and stuff. This model shows 61.4% according to the PQ metric on the Citysacpes dataset.

Despite the wide coverage in the scientific literature of the problems of segmentation of road objects, almost no attention has been paid to the use of segmentation for the automatic construction of 3D scenes of road accidents. This, together with insufficient general automation of building 3D accident scenes, including the selection of objects and estimating the distances between them, actualizes the problem of using Deep Learning Segmentation methods to solve the problem of automatically constructing a 3D accident scene.

The purpose of this work is to automate the process of recreating an accident scheme by developing an autonomous system for collecting and analyzing data of an accident site and then providing an accident scheme with detected road infrastructure objects (marking, signs, fences, etc.), recognized vehicles. The novelty of the work lies in the use of the well-known Panoptic Segmentation neural networks, trained on the Cityscapes dataset, to segment accident objects during the construction of a 3D accident scene using UAV, as well as in clustering the obtained 3D scene textures to detect objects on a 3D scene and the method for automatically determining the distance between objects, which consists in calculating the distance between clustered objects as a minimum of Euclidean distances between all points of equally clustered objects.

Recently, studies have been conducted on the recognition of volumetric objects in a flat image. The CubeSLAM algorithm solves the problem of detecting a three-dimensional object on a flat image from a monocular camera and aligns (optimizes) the position of this object and the position of the camera [41]. The system is based on the key points (features) of ORB SLAM2 [42], which includes an external camera tracking interface and an internal Bundle Adjustment (BA) interface. BA is a process of collaborative optimization of various map components, such as poses and camera points [42,43]. To associate objects, CubeSLAM uses a feature-point matching method. First, feature points are associated with their corresponding object (Figure 1) if the points are observed in the bounding box of the 2D object for at least two frames, and their three-dimensional distance to the center of the defining cuboid is less than 1 m. Then, two objects in different frames are associated if they have the greatest number of common characteristic points among themselves, and this number must also exceed a certain threshold (10 points).

To detect 2D objects, CubeSLAM uses the YOLO detector [45] with a probability threshold of 0.25 for indoor scenarios and MS-CNN [46] with a probability of 0.5 for outdoor testing. Both detectors run in real time on the GPU.

In scenarios where precise camera positioning is not provided, the algorithm samples the camera roll/pitch within a range of ±20° around the originally estimated angle.

One of the advantages of CubeSLAM is that it does not require big data for training, since only two weights need to be configured. CubeSLAM can also work in real time, including 2D object detection and mapping.

However, this algorithm does not guarantee an autonomous reconstruction of an accident scheme, and the authors do not provide a methodology for collecting data and do not provide a highly accurate determination of the distances between objects on the diagram. In order to achieve this goal, the following methods were used in this work: methods for constructing UAV motion trajectories to collect an extended set of data about the scene of an accident, deep learning methods for detecting voluminous objects based on a three-dimensional model created based on the collected data, image transformation methods for constructing a two-dimensional scheme, methods for comparing recognized objects in a three-dimensional model and a flat two-dimensional image, as well as methods for accurately calculating the distances between objects in a road accident scheme.

## 2. Materials and Methods

This section describes in more detail all of the methods and algorithms used.

### 2.1. Planning of the UAV Trajectory for the Survey and Collection of Data on Road Accidents

Two trajectories of movement can be used to collect data on road accidents by means of UAV: circular and covering a given area. The first type of trajectory (Figure 2a) is implemented in a lightly loaded environment. To build it, the operator of the field service on the map determines the center of the circle and the radius. The UAV moves along the constructed trajectory so that the camera is always directed to the specified center of the circle. The second type of trajectory (Figure 2b) is more flexible in construction. The operator selects an area of interest, which can have any geometric shape, camera parameters, and frame overlap percentage. The UAV onboard computer calculates a trajectory that evenly covers a given area, not including the marked areas of potential obstacles.

In Figure 2, the green polygon marks the area explored by the UAV, and the white line shows the planned trajectory of the UAV. The bright green point marks the take-off coordinate of the UAV, and the orange line marks the flight trajectory to the specified area and the return trajectory.

To determine the appropriate UAV flight path for information gathering purposes, a series of experiments were conducted on a sample of 50 accident scenes in the simulator. Simulation in Gazebo was carried out with standard UAV quadcopters. The used UAV “iris” has the following characteristics: dimensions 440 mm × 330 mm × 100 mm, weight 1.2 kg, and maximum speed 18 m/s. A camera with a pixel count of 24.3 MP, an APS-C type Exmor CMOS sensor of 23.5 × 15.6 mm, and an overlap percentage of 25% is installed on board the UAV.

The flyby trajectory of the first type (Figure 2a) is built automatically after entering the coordinates of the intended center of the scene, radius, and height of the flyby. While moving along the trajectory of the first type, the UAV takes photos with a given parameter of frame overlap. This is required to create an accident data collection. To calculate the location of each waypoint where the photograph will be taken, it is necessary to calculate the distance between the previous and current UAV positions, i.e., step size. The step size depends on the required distance to the object of interest (GSD [cm/pixel]) and the overlap parameter (overlap) (Figure 3).

The frame has a width ImW and a height ImL of the image in pixels. Knowing these values, it is possible to calculate the step length in meters along the frame width p1 and along the frame height p2 using Formulas (1) and (2):(1)$$p1=\frac{ImW\times GSD}{100}\times \left(1-overlap\right),$$
(2)$$p2=\frac{ImL\times GSD}{100}\times \left(1-overlap\right),$$
where GSD is calculated by Formula (3):(3)$$GSD=\frac{H\times Sw\times 100}{F\times ImW},$$
where H is the UAV flight altitude, Sw is the width of the matrix (camera sensor), and F is the focal length.

Longitudinal pitch (p1) and transverse pitch (p2) provide the specified overlap value. In Figure 3, three areas are highlighted, represented by green, yellow, and red colors, corresponding to different acquired images and their overlap zone, determined by the length and width of the frame, and the specified overlap value. The colored dots represent the subsequent positions of the UAV from which the three images were acquired.

When the UAV moves along an elliptical trajectory describing the object of interest (Figure 4), the rotation of the stabilizer with the camera ϑ is calculated by the following Formula (4):(4)$$\vartheta =\tan^{-1}\frac{\rho }{H-h_{obj}}$$
where ϑ is the distance to the object, $h_{obj}\;$is the height of the object of research.

The trajectory of the coating of the second type (Figure 2b) is a particular problem in robotics. The solution to the CPP problem is the planned trajectory of the robot. To solve the CPP problem, methods are used to determine the path to completely cover a certain area. To solve the problem of constructing a trajectory on a regular-shaped section without obstacles using one UAV, simple geometric patterns of movement are sufficient. The most common patterns are the back-and-forth and the spiral [47]. To create a reciprocating trajectory, one needs to know that d is the distance (Figure 5) between two rows. The calculation of this distance depends on the specified vertical overlap and camera parameters.

Let v be the vertical overlap and w be the width of the camera frame. The row spacing, denoted as d, is the (vertical) distance between two frames. Taking into account the vertical overlap, d is calculated as follows:(5)d=w·(1−v)

The number of turns n that must be performed in this polygon depends on the values of d*,*
w, and ls is the length of the optimal line sweep direction. It is important to define an intermediate value z=ls−w/2, where w/2 represents half the size of the camera coverage area. Graphically, these parameters are presented in Figure 6.

Let d define the distance separating the rows. [z/d] denotes the number of rows required to cover the polygon using the reciprocating pattern. However, if the distance between the last row and the top vertex (z mod d) is greater than w/2, the polygon is not completely covered by the last row and another row is required.

Each segment requires two pivot points, resulting in a total number of pivots given by:(6)$$n=\{\begin{array}{c} 2\;\cdot \left[z\backslash d\right],\;if\;z\;mod\;d\;\leq w/2\\ 2\;\cdot \left(\left[z\backslash d\right]+1\right),\;if\;z\;mod\;d>w/2 \end{array}$$

The number of rotations depends on z, since d is given when setting the problem. The parameter z also depends on the length of ls.

The following parameters were set for the movement of the UAV in the simulator: the average speed of the UAV movement was 3 m/s and the flight altitude was 8 m–10 m. As a result, the survey area for the trajectory of the first type turned out to be 95 m^2^, and for the trajectory of the second type is 100 m^2^.

According to the results of the experiments, it was found that the average time of movement along the calculated trajectory of the first type, taking into account the time of takeoff and landing, is 21.1 s, which is an order of magnitude less than the average time of movement along the trajectory of the second type is 52.5 s. This also indicates that less energy was used for flight along the trajectory of the first type than for the second type. After flying along the trajectory of the first type, the UAV had, on average, 91% of the battery charge, while after flight along the trajectory of the second type, the UAV had, on average, 83%. An assessment was also made of the quality, reliability, and completeness of the collected photo and video material about the scene of an accident. As a result, it was possible to establish that the trajectory of the first type provides the necessary and sufficient data for a more accurate reconstruction of the terrain in a short time compared to the trajectory of the second type.

### 2.2. Method for Detecting Key Objects of an Accident

Since it is necessary to segment both background stuff and a number of distinct things in UAV images, the panoptic segmentation approach is used in this article. To perform panoptic segmentation of road accident scenes, the deep convolutional model Switchable Wide Residual Network (SWideRNet) with Axial Attention [48] was used, which was previously trained on the panoptic segmentation task on the Cityscapes dataset. It is important to take a closer look at this architecture. The choice of this neural network architecture was due to the results shown in the general problem Panoptic Segmentation of the road traffic objects on the Cityscapes dataset [48]. In addition, within the framework of this work, a comparative study of the results of segmentation was performed on the data sample prepared by us from other Panoptic Segmentation architectures: Panoptic Deeplab [38], Axial Deeplab [49], MaxDeeplab [50]. The results are shown in Section 4.3.

This architecture generally implements the encoder–decoder architecture and is based on the Panoptic Deeplab architecture. The encoder extracts feature maps from the input images. Then the extracted feature maps are fed into the decoder, which generates both the segmentation map of stuff and the segmentation maps with the individual things. The Panoptic Deeplab architecture is presented in Figure 7.

This neural network implements panoptic segmentation as follows: semantic segmentation of stuff is performed in parallel with instance segmentation of things. Semantic segmentation is performed using the Deeplab architecture, the main feature of which is the use of the so-called Atrous Convolutions [23] and Spatial Pyramid Pooling in the decoder part. Instance segmentation task, in turn, is subdivided into two tasks: instance center prediction and instance center regression. The results of these two procedures are grouped in the following. The final panoptic segmentation map is generated by fusing semantic segmentation map and groped instance prediction and regression results.

The other distinctive features of SWideRNet are Wide Residual Blocks with Switchable Atrous Convolution (SAC) [51] and Axial-Attention blocks [49].

Wide residual block (WideResNet) [52] is the modification of plain residual block [29]. Wide residual block differs from plain residual block by the increasing of the channel number and decreasing the depth of neutral networks. Wide Residual Blocks with SAC in turn have additional neural structures, which are depicted in Figure 8.

The first unit in the depicted block is SAC. The main distinctive feature of it is the switching of two parallel branches: the branch with atrous convolution and the branch with plain convolution. The routing procedure is performed by multiplication with the hard feature maps sigmoid output of the middle branch of the SAC block. The second special unit of the Wide Residual Blocks with SAC is the simplified Squeeze-and-Excitation unit [53], which consists of a fully connected layer, followed by hard sigmoid activation.

Axial Attention block processes feature maps as follows. At the first stage, multihead self-attention [54] processes only columns of the feature maps. At the second stage, multihead self-attention processes the rows of the feature maps obtained from the column-wise multihead self-attention. In SWideRNet architecture, Axial blocks are incorporated into the residual structure. Residual Axial Attention are only used only at the two last stages in the neural network decoder. Residual Axial Attention is depicted on Figure 9.

In this work, to perform segmentation, the DeepLab2 implementation [55] was used, which was implemented using the TensorFlow library [56] and contains the SWideRNet model, which was trained on the Cityscapes dataset. The complete list of classes represented in this dataset is as follows: road, sidewalk, building, wall, fence, pole, traffic light, traffic sign, vegetation, terrain, sky, person, rider (of bicycle or motorcycle), car, truck, bus, train, motorcycle, and bicycle. Thus, within the framework of this work, only those objects that are included in the set of Cityscapes classes are automatically segmented.

### 2.3. Method for Automatic Measurement of Distances between Detected Objects of an Accident

The resulting segmented images contain homogeneous regions of pixels (i.e., having the same color) belonging to the same class. This homogeneity can be used to calculate the size of objects (in this article, these are cars), as well as to calculate the calculation of distances between objects (distances from cars to sidewalks).

Consider, first, the approach used to calculate the size of segmented cars. Figure 10 shows an example of a segmented car.

A homogeneous area of a segmented car can be represented as a matrix of coordinates of its pixels:(7)$$X=\left[\begin{array}{ccc} \begin{array}{c} c_{1}\\ r_{1} \end{array} & \begin{array}{c} \ldots \\ \ldots \end{array} & \begin{array}{c} c_{N}\\ r_{N} \end{array} \end{array}\right]=\left[\begin{array}{c} c\\ r \end{array}\right]$$
where $c_{i}$ is a column coordinate of *i*-th pixel, $r_{i}$ is a row coordinate of *i*-th pixel, and N is a number of pixels that correspond to a segmented area. To calculate the coordinates of the $v_{1}$ and $v_{2}$ axes, along which the segmented area is rotated, it is necessary to perform Principal Component Analysis (PCA) [57] of the matrix **X**:(8)$$XX^{T}=V\Lambda V^{T}$$
where $V=\left[\begin{array}{cc} v_{1} & v_{2} \end{array}\right]$ is a matrix, which columns are principal directions or eigenvectors, along which the segmented area is rotated; Λ is a diagonal matrix with eigenvalues of the corresponding eigenvector.

Then, the projection of X into the eigenspace V is performed:(9)$$X^{p}=VX^{T}=\left[\begin{array}{ccc} \begin{array}{c} c_{1}^{p}\\ r_{1}^{p} \end{array} & \begin{array}{c} \ldots \\ \ldots \end{array} & \begin{array}{c} c_{N}^{p}\\ r_{N}^{p} \end{array} \end{array}\right]=\left[\begin{array}{c} c^{p}\\ r^{p} \end{array}\right]$$
where $c_{i}^{p}$ is a column coordinate of *i*-th pixel projected to the eigenspace, $r_{i}^{p}$ is a row coordinate of *i*-th pixel projected to the eigenspace, and N is a number of pixels that correspond to a segmented area.

The length of the segmented vehicle is calculated as follows:(10)$$l=\max\left(c^{p}\right)-\min\left(c^{p}\right)$$

The width of the segmented vehicle is calculated as follows:(11)$$w=\max\left(r^{p}\right)-\min\left(r^{p}\right)$$

On the contrary, the same pixel values of the segmentation maps, the homogeneity of pixel regions makes it possible to calculate the distances between these regions. Figure 11 schematically shows the procedure for calculating the distances between areas belonging to different objects.

This procedure includes the following steps:All closed areas on images that have the same color are determined;The Euclidean distance of all pixels of each area with the boundary of other segmented areas is calculated;The distance between areas belonging to different objects that do not border on each other is determined at least from all calculated distances between the boundary pixels of these objects.

## 3. Suggested Approach to Road Accident Mapping

To implement the autonomous construction of a road accident map, it is necessary to combine all the previously described methods into a single approach. Each step is shown in turn in Figure 12. The first stage is the collection of data about an accident scene. Data collection is carried out by means of the UAV. To collect enough data on an accident, the coordinate of the center of the examination scene, the radius for constructing the trajectory of the first type, and the percentage of overlapping images are sent to the UAV memory.

All intermediate results obtained during the processing of a set of images of an accident scene are shown in Figure 13.

Next, the data set is processed by the SWideRNet neural network, which generates panoptic segmentation maps. These generated maps contain segmented small objects (things) and large background objects (stuff). The difference of things from each other by color allows one to measure the distances between objects.

After obtaining a segmentation map of each image, this map is superimposed on the original image. Overlaying one image on another is based on the use of alpha channels and the Pillow library. The Pillow Python image processing library provides alpha blending for images using the Image class’s alpha_composite() method. In computer graphics, alpha blending is the process of combining a single image with a background to create the effect of partial or complete transparency. This is necessary in order to render image elements (pixels) in separate passes or layers and then combine the resulting 2D images into one final image, called a composite. Transparency levels are determined by the pixel color of both images in the same way:(12)$$I_{i,j}^{out}=\left(1-\alpha \right)I_{i,j}^{in}+\alpha I_{i,j}^{mask}$$
where α is a blending coefficient (in this work α=0.55), i is a row index of an image pixel, j is a column index of the image pixel, $I^{in}$ is the image from the UAV, $I^{mask}$ is the panoptic segmentation mask of the $I^{in}$ image, and $I^{out}$ is the alpha-embedded image.

The construction of a three-dimensional model of an accident is based on the combined source images and the corresponding segmentation map. This allows one to “color” the recognized road objects on the reconstructed three-dimensional model, which is subsequently necessary to build a road accident map and calculate the distances between the recognized road objects.

A 3D accident model is an intermediate result of processing the entire set of data about a scene with an accident. The stage of building an accident model is necessary for the automated calculation of distances between key objects that were recognized at the second stage.

The resulting 3D model has textures colored in the colors of the segmentation maps. However, due to the alpha blending of the segmentation map into the original image from the UAV, as well as the peculiarities of methods for constructing 3D models based on images using photogrammetry, the resulting 3D model textures have an irregular structure in terms of color filling of areas belonging to different image classes. To ensure greater homogeneity of segmented textures, additional clustering of texture pixels is performed using the k-means clustering algorithm [58,59] with a preliminary number of clusters equal to 20.

The next step is to obtain an output image from the 0XY plane of the 3D model (Figure 14).

For this, a transformation using a projection matrix is used. It displays the relationship between the pixel (x, y) of the plan image and the pixel (u, v) of the input image.
(13)$$\left[\begin{array}{c} x\prime \\ y\prime \\ w\prime \end{array}\right]=\left[\begin{array}{ccc} a_{11} & a_{12} & a_{13}\\ a_{21} & a_{22} & a_{23}\\ a_{31} & a_{32} & a_{33} \end{array}\right]\left[\begin{array}{c} u\\ v\\ w \end{array}\right],\;\mathrm{where}\;x=\frac{x\prime }{w\prime }\;\mathrm{and}\;y=\frac{y\prime }{w\prime }.$$

The transformation is commonly referred to as inverse perspective mapping (IPM) [60]. The IPM takes a front view as input, applies homography, and creates a top-down view of the captured scene by mapping pixels to a 2D frame (bird’s-eye view). In practice, IPM works perfectly in the immediate vicinity of the car and provided that the road surface is flat [60].

Based on the obtained top view, the distances between objects are calculated in the manner described above in Section 2.3.

## 4. Experiments

### 4.1. Building a 3D Scene

The effectiveness of the proposed approach was tested on a sample of 20 accident scenes. Every scene was created in Gazebo. Scenes were generated using objects such as roads, intersections, poles, buildings, trees, cars of different sizes and colors, road signs, and pedestrians. Some models (buildings, trees, cars) of objects were taken from the Gazebo library of standard models, while some were created independently (road, road signs, road markings). When creating a selection of scenes, references to real-life intersections and other road sections were used. The Gazebo simulator is convenient when generating a large number of scenes with many small objects. In addition, it is easy to implement a UAV model flight in Gazebo for the purpose of collecting information. Simulation in the Gazebo was carried out with standard UAV quadcopters. The used UAV “iris” has the following characteristics: dimensions 440 mm × 330 mm × 100 mm, weight 1.2 kg, maximum speed 18 m/s. To collect data on each scene, one must specify the coordinate of the center of the examination scene, the radius for constructing the trajectory of the first type, and the percentage of image overlap. The data set for each accident scene consisted of at least 50 images, which allows one to accurately recreate the three-dimensional accident scene.

An example of the original scenes with measured reference distances and their corresponding generated 3D models are shown in Figure 15.

As a result, a sample consisting of 20 scenes was compiled, based on the images of which the corresponding 3D models were generated. All 3D models were built using the open-source photogrammetric framework AliceVision. A preliminary analysis showed that out of the entire Cityscapse dataset on 3D models only objects of classes are adequately displayed: vehicle, road, and sidewalk. In addition, in the case of an accident, we are primarily interested in the distance between vehicles, as well as the distances from vehicles to areas prohibited for traffic (sidewalk, building, terrain). Therefore, within the framework of this article, only the above classes will be considered. Figure 16 shows a 3D model obtained using photogrammetry.

Figure 17 shows a top view of the constructed 3D scene (a) a clustered top view image with calculated distances between objects (b).

### 4.2. Metrics

To evaluate the results of object segmentation, the Intersection over Union (IoU) metric was used:(14)$$IoU=\frac{\left|S_{true}\cap^{}S_{pred}\right|}{\left|S_{true}\cup^{}S_{pred}\right|}$$
where $S_{true}$ is a true area and $S_{pred}$ is an automatically segmented area.

To assess the accuracy of calculating the distances between key objects, it is necessary to compare the real distances between the models on the scene and the obtained distances between the same objects, but on the recreated 3D model. This comparison is drawn by calculating the absolute error between the reference distance and the automatically calculated distance:(15)$$err_{i}=\left|d_{i}-{\hat{d}}_{i}\right|$$
where $err_{i}$ is the calculated error of the i-measurement of the distance, $d_{i}$ is the reference value of the distance, and ${\hat{d}}_{i}$ is the value of the distance calculated automatically by the method given in this article.

To estimate the measurement error for the entire sample, its confidence interval was calculated:(16)$$MAE=\bar{MAE}\pm \frac{z_{c}s}{\sqrt{n}}$$
where n is the sample size, $\bar{MAE}=\frac{1}{n}\sum_{i=1}^{n}err_{i}$ is the sample mean error, $s=\sqrt{\frac{1}{n-1}\sum_{i=1}^{n}{\left(err_{i}-\bar{MAE}\right)}^{2}}\;$is the sample standard deviation, and $z_{c}$ is the value of the coefficient of two-tailed z-test for the specified confidence level c, which in this work was chosen equal to c=0.99.

### 4.3. Results and Discussion

As part of this work, IoU was calculated both for the entire sample and for each class separately. The results of the IoU calculation on the sample are summarized in Table 1.

Table 1 shows that the best IoU performance was achieved by SWideRNet with Axial Attention both for all classes as a whole and for each class separately. It is also worth noting that with sufficiently high IoU rates when segmenting the roadway, sidewalks and the average IoU for all classes, the proposed approach does not allow the achievement of high vehicle segmentation results. This is explained by the heterogeneity of textures in the formation of a 3D scene using photogrammetry, which, together with the formation of the final segmentation maps using clustering, leads to numerous artifacts in the resulting segmentation maps.

Within the framework of this research, the width of the road (distance between sidewalks), the distance between vehicles and sidewalks, and the distance between vehicles were measured. As a result, a total of 148 distances were measured in a sample of 20 scenes.

As a result of sample processing, error values for each model were obtained, taking into account the confidence interval (confidence level is 0.99; see Formula (16)), summarized in Table 2.

The table shows that the best (minimum) MAE values were achieved with SWideRNet with Axial Attention. It is also worth noting that the resulting error result, taking into account the confidence interval, allows us to conclude that there are acceptable discrepancies between the measured and reference sizes. It should also be noted that a low distance measurement error is achieved under conditions of relatively low vehicle segmentation results (IoU=0.484).

## 5. Conclusions

The approach presented in this article makes it possible to automate the construction of an accident scheme: the scheme itself is constructed; in this scheme, the areas belonging to various objects are determined, while the distances between objects on the generated scene are automatically measured. The above approach showed good results in detection by means of segmentation of objects of interest (mIoU = 0.771). The analysis of confidence intervals showed a low measurement error in measuring distances between objects (MAE = 0.142 ± 0.023 m). The approach presented in this article can also be applied to automatically construct real accident scenes.

In future work, we plan to develop our research in the direction of performing all processing on the UAV, the computing power of which may not be enough to process deep neural networks that perform segmentation of 3D objects. Nevertheless, as part of further work, it is also planned that 3D scene segmentation models will be tested. At the moment, we are working on this: we are collecting a dataset from aerial photographs of more than 100 scenes of real accidents for further training of our system and further testing of the proposed method on real data.

## Figures and Tables

**Figure 1 sensors-22-04728-f001:**
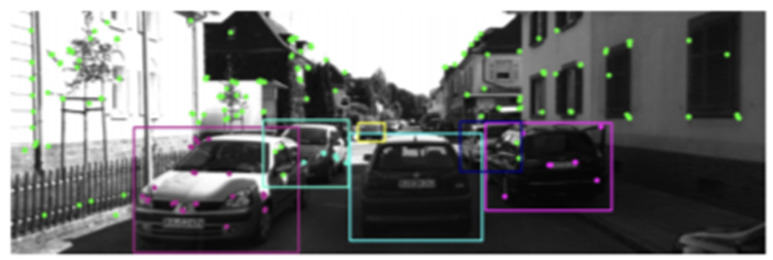
Association of objects in dynamic and occluded scenarios KITTI 07 [44]. Green points are non-objective points, points of other colors are associated with objects of the same color. The front blue moving car is not added as a SLAM landmark since no feature point is associated with it since the car is in motion. Points in the feature overlap region are not associated with any feature due to ambiguity (belonging to two features).

**Figure 2 sensors-22-04728-f002:**
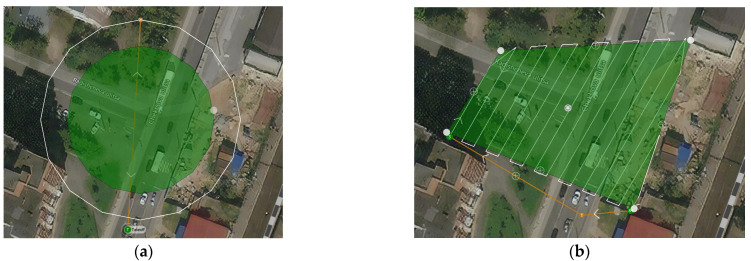
Examples of trajectories for the reconstruction of an accident: (**a**) shows the first type of trajectory; (**b**) the second type of trajectory.

**Figure 3 sensors-22-04728-f003:**
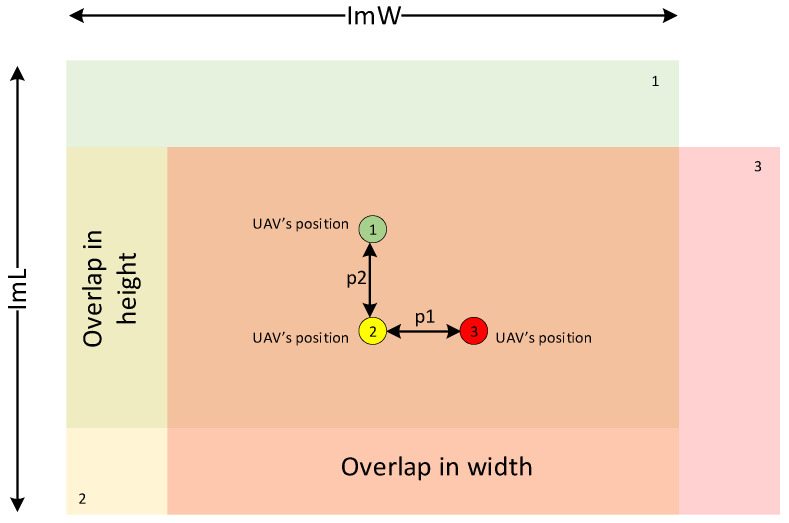
Demonstration of frame overlap in width and height. Each color corresponds to a camera frame: green is the first frame, green circle with number 1 is the center of this frame; yellow color is the second frame, yellow circle with number 2 is the center of this frame; red is the third frame centered on a red circle with the number 3.

**Figure 4 sensors-22-04728-f004:**
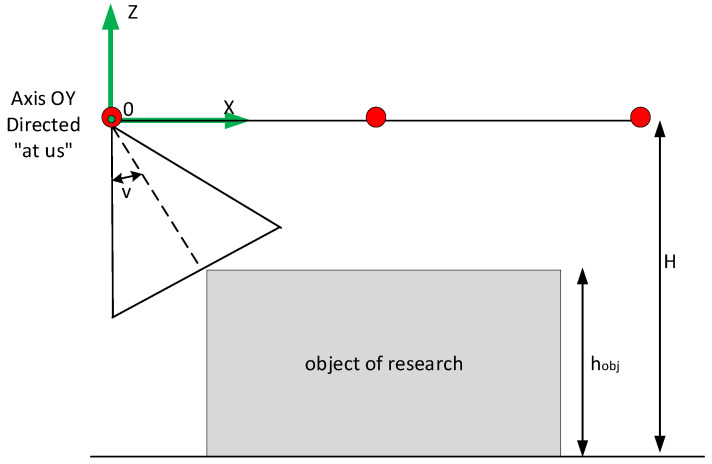
Illustration relating the parameters used in the formulas for calculating the rotation of the stabilizer. The red dots are the positions of the camera in space.

**Figure 5 sensors-22-04728-f005:**
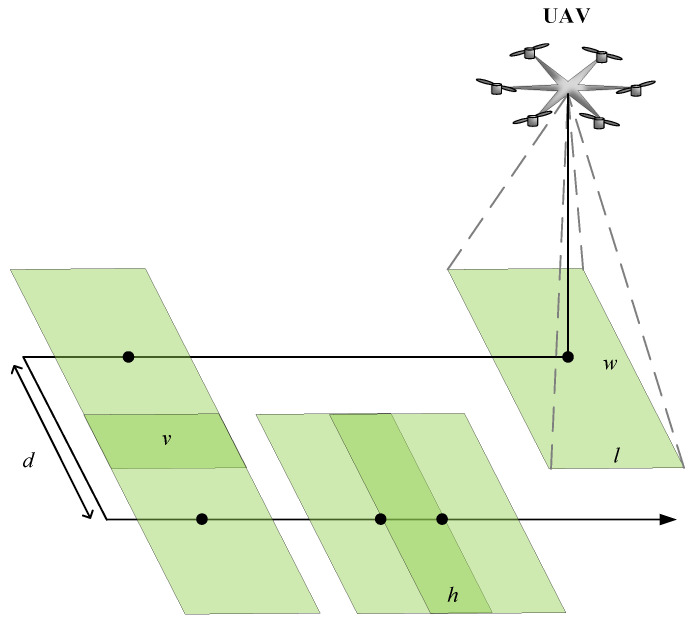
Camera parameters: length l and width w. Overlap Options: h and v denote the percentage of horizontal and vertical overlap of images. Each new frame is marked in green.

**Figure 6 sensors-22-04728-f006:**
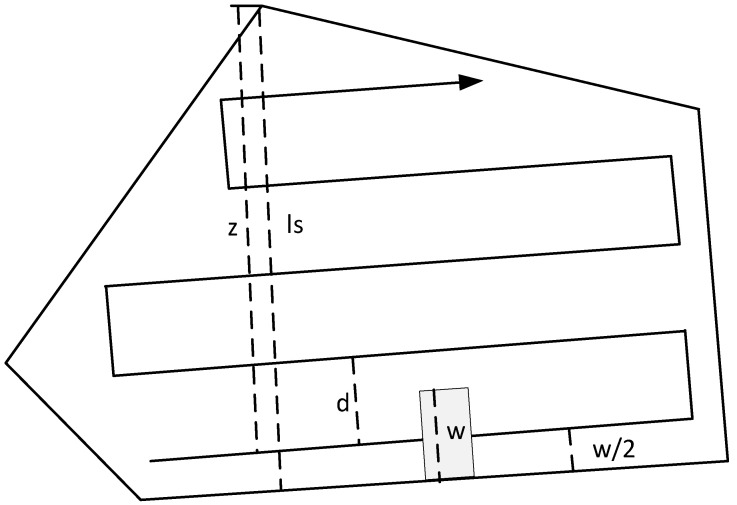
Image of the values needed to calculate the number of turns.

**Figure 7 sensors-22-04728-f007:**
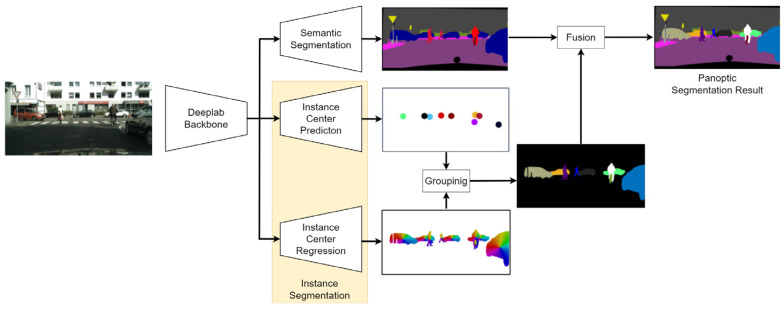
Overall Panoptic Deeplab architecture.

**Figure 8 sensors-22-04728-f008:**
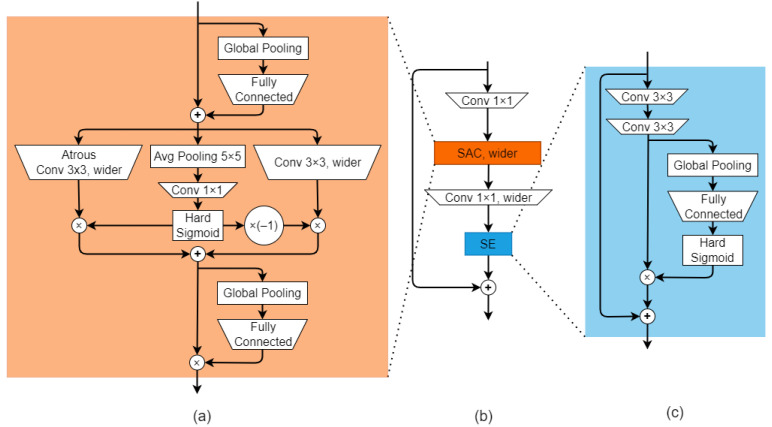
Wide Residual Blocks with Switchable Atrous Convolution: (**a**)—SAC unit, the main disitinctive feature of which is the switching of two parallel branches: the branch with atrous convolution and the branch with plain convolution. The routing procedure is performed by multiplication with the hard feature maps sigmoid output of the middle branch of the SAC block; (**b**)—an overall of Wide Residual Blocks with Switchable Atrous Convolution; (**c**)—simplified Squeeze-and-Excitation unit [53], which consists of a fully connected layer, followed by hard sigmoid activation.

**Figure 9 sensors-22-04728-f009:**
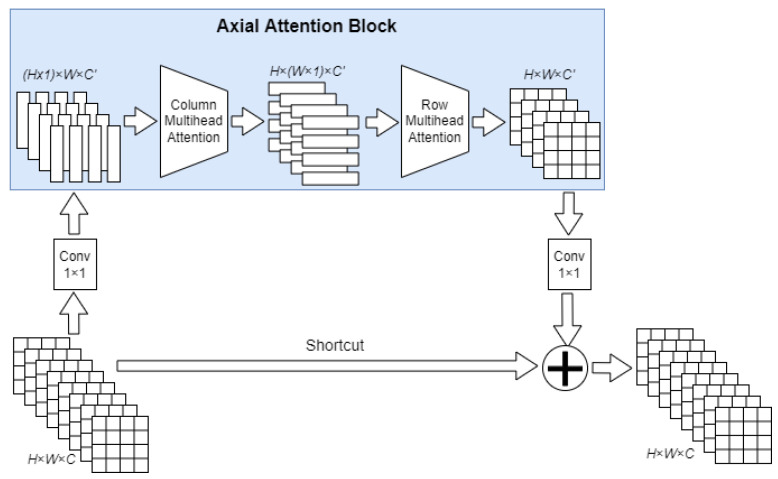
Residual Unit with Axial-attention block.

**Figure 10 sensors-22-04728-f010:**
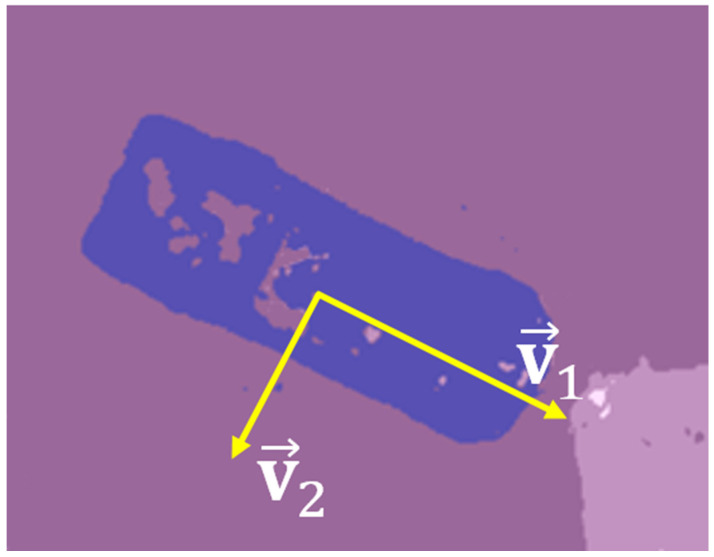
Segmented car. The car has an elongated shape and is rotated along axes $v_{1}$ and $v_{2}$.

**Figure 11 sensors-22-04728-f011:**
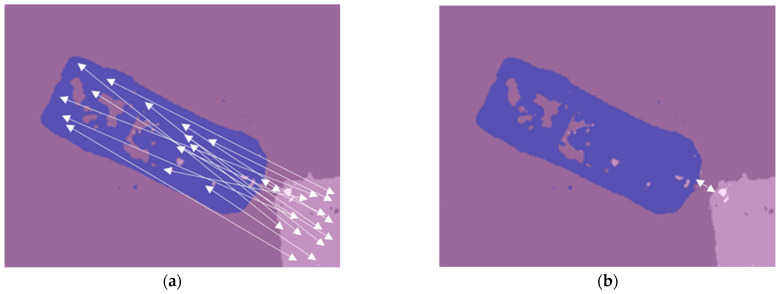
Automatic calculation of distances between objects: (**a**) distances between pixels belonging to different classes (only 15 distances are shown for readability); (**b**) minimum distance. The white arrows illustrate the distances between pixels belonging to different classes.

**Figure 12 sensors-22-04728-f012:**
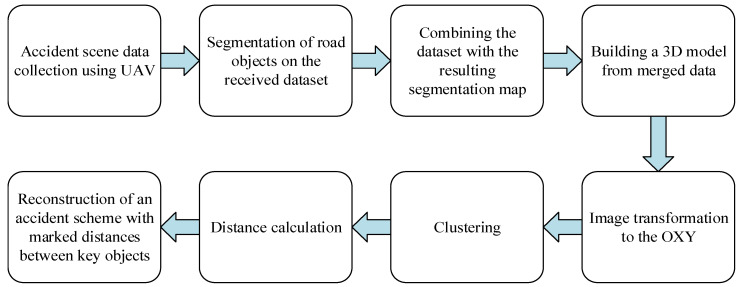
Stages of the approach to the autonomous construction of a road accident map.

**Figure 13 sensors-22-04728-f013:**
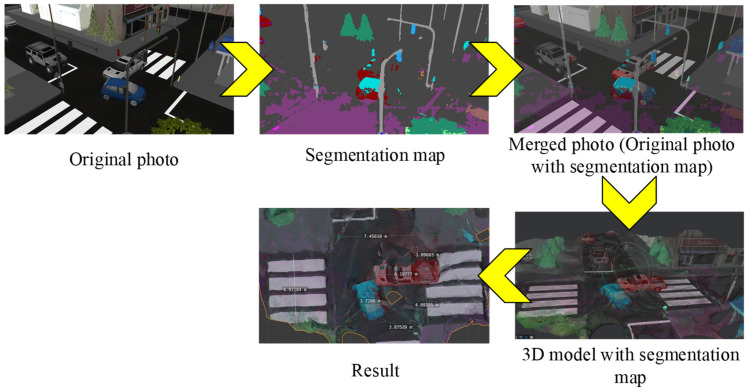
Intermediate results obtained during image processing and construction of an accident map. Each color is responsible for belonging of the detected object to a certain class. For example, green is trees and other vegetation, purple is a road, red is one car, blue is another.

**Figure 14 sensors-22-04728-f014:**
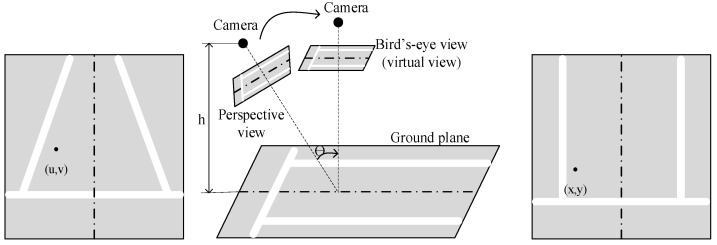
Demonstration of image transformation.

**Figure 15 sensors-22-04728-f015:**
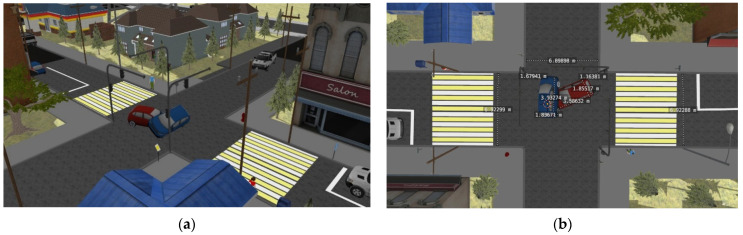
Scenes in Gazebo containing accidents and reference distances between objects: (**a**) illustrates a model of an accident scene, (**b**) shows hand-measured distances in a photograph of an accident scene.

**Figure 16 sensors-22-04728-f016:**
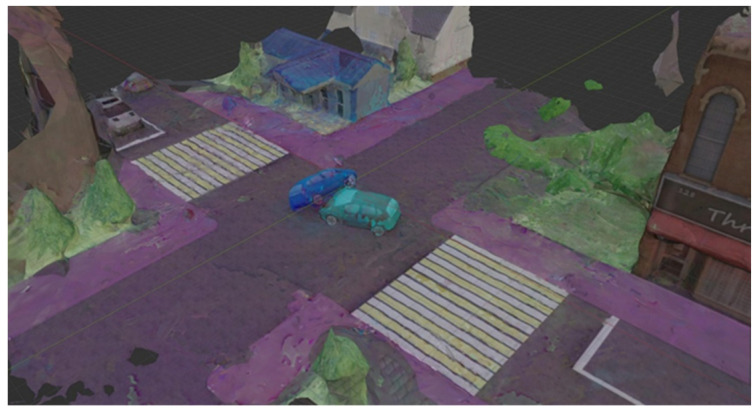
3D scene. With photogrammetry based on segmented images, the resulting textures contain information about segmented objects in the form of characteristic colors.

**Figure 17 sensors-22-04728-f017:**
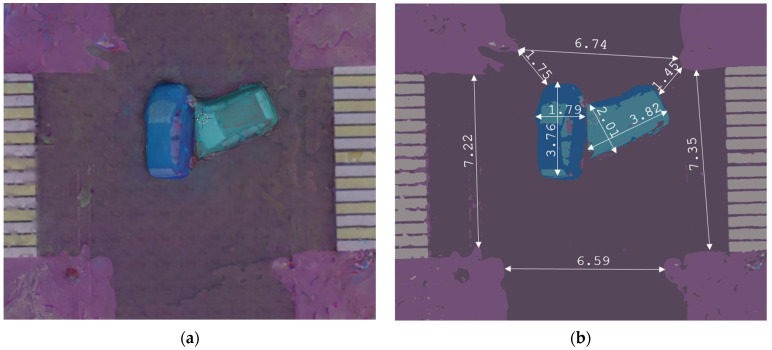
Top view of the 3D scene (**a**); clustered top view with calculated distances between key objects (**b**).

**Table 1 sensors-22-04728-t001:** IoU results.

Model	Vehicle	Road	Sidewalk	Mean
Panoptic Deeplab [48]	0.423	0.797	0.822	0.721
Axial Deeplab [49]	0.479	0.829	0.865	0.762
Max Deeplab [50]	0.465	0.812	0.874	0.753
SWideRNet with Axial Attention [51]	**0.484**	**0.839**	**0.892**	**0.771**

**Table 2 sensors-22-04728-t002:** Error values for measuring distances between objects on the scene of an accident.

	Panoptic Deeplab [48]	Axial Deeplab [49]	Max Deeplab [50]	SWideRNet with Axial Attention [51]
Measurement error, m	0.202 ± 0.044	0.192 ± 0.022	0.163 ± 0.032	**0.142 ± 0.023**

## Data Availability

Not applicable.

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
