# Peer review of "An Approach to the Automatic Construction of a Road Accident Scheme Using UAV and Deep Learning Methods"

_sensors, 2022, doi:10.3390/s22134728_

Round 1

Reviewer 1 Report

The paper has described a method of using UAV for traffic accident scene reconstruction. The deep learning-based method is used to capture 3-D scene elements from the crash site. The following comments are given to the authors:

  1. The reconstruction of the traffic accident scene is an interesting and practical relevant topic. However, the authors haven't really explained the background very well. Some transportation safety-related elements must included in the background, so that we know what are the key elements in these sort of accident scene. Also what are the exact application of this proposed method. is it for traffic safety related study or accident scene investigation. Authors should clarify this to readers.
  2. In the urban environment, most of traffic accidents are quite minor. Usually it happens during the peak hour, accessing to the accident scene is difficult.  Therefore we can use UAV to capture the evidence from the scene and drive away the vehicles. Police or the insurance company will collect the evidence afterwards. In this case, there is a clear motivation behind it. This paper's motivation is not that clear. If the police is already there, why don't just take the photo or video themselves, why bother using UAVs?
  3. The method employed in this paper is quite traditional and well-established methods. The contribution of this paper is not very clear. 
  4. Also does UAV's battery constraint consider in the UAV path planning?
  5. Formulas in the paper are not numbered. 
  6. The novelty of the paper needs to be clearly pointed out. Currently, i'm afraid i don't see any novelty in this paper. 

Author Response

Thanks for your feedback and recommendations! They are very important to us, because in this way we can do our work much better. Let us answer your comments.

  1. Despite the many programs and technologies already implemented to collect data on the location of road traffic accidents, the number of such accidents recorded worldwide is still growing and, accordingly, the burden on services to provide analysis of the causes of accidents is growing. So, our work is related to the investigation of the scene and the analysis of data on a road accident. We will definitely add this to the introduction so that the reader is clear about what the text is talking about.
  2. The police are not always the first to arrive at the scene of a traffic accident. However, data collection and road mapping is currently mostly a manual procedure performed by people. The accuracy of the map of the scene of a traffic accident depends mainly on the human factor. We see it like this: the use of UAV is necessary when it is not possible to collect sufficient and reliable data quickly. If drivers take pictures of the incident, this data will not be enough to establish the cause of the accident and search for the involvement of drivers. Autonomous UAV allows you to capture the entire scene of an accident completely, and methods for recognizing and calculating the distances between key objects guarantee high-precision analysis of the received data. It is assumed that the user, after he is convinced that none of the participants in the accident does not need medical assistance, reports the incident to the police and the police send an equipped UAV from the nearest station/office to the scene. The UAV autonomously collects data and returns to base. If the accident is not serious, no one was hurt, the drivers can leave, and the po-lice begin processing the application and the received data. Then they send the final decision on the cause of the accident to all participants in the accident and their insurance companies.
  3. The novelty of the study is justified primarily by the fact that in modern scientific literature the issues of automatic segmentation of objects in the construction of a 3D scene of an accident are insufficiently covered. The issues of automatic measurement of the distance between objects are also insufficiently covered. In well-known works, despite the automatic construction of 3D scenes, the selection of objects in the 3D scene and the measurement of all distances between objects is performed manually. We have revealed the rationale for novelty by expanding the review of sources.
  4. Yes, of course it is important. In the new version of the document, we have presented quantitative results on the remaining charge of the battery in our experiments.
  5. We took into account your comment and corrected it in the new version of the document.
  6. The novelty of the work lies in the use of the well-known Panoptic Segmentation neu-ral networks, trained on the Cityscapes dataset, to segment accident objects during the construction of a 3D accident scene using UAV, as well as in clustering the obtained 3D scene textures to detect objects on a 3D scene and the method of automatic distance determination. between objects, which consists in calculating the distance between clustered objects as a minimum of Euclidean distances between all points of equally clustered objects.

We hope that we have fully answered all your questions and comments.

Thank you for your cooperation!

Reviewer 2 Report

1.Key references are missing: Research on traffic accident prediction and object detection from aerial perspective should be paid attention and introduced. e,g,

Najjar, A., Kaneko, S., & Miyanaga, Y. (2017). Combining Satellite Imagery and Open Data to Map Road Safety. Proceedings of the AAAI Conference on Artificial Intelligence, 31(1). 

Z. Wei et al., "Learning Calibrated-Guidance for Object Detection in Aerial Images," in IEEE Journal of Selected Topics in Applied Earth Observations and Remote Sensing, vol. 15, pp. 2721-2733, 2022

2.The quantitative experimental results are missing, please supplement the comparative experiments of the methods of the peers, preferably on the public datasets or on the datasets prepared by yourself.

3.In Fig7, Overall Panoptic Deeplab architecture. Why are the results before and after fusion the same? Please explain.

The innovation of this paper is the use of semantic segmentation for automatic reconstruction of traffic accidents. I don't think the improvement of semantic segmentation models is the point. If yes, please supplement the ablation experiment results. Please further highlight the contribution of this article.

Author Response

Thanks for your feedback and recommendations! They are very important to us, because in this way we can do our work much better. Let us answer your comments.

  1. Yes, you are right. We added key references to the new version of the document.
  2. Of course, we will take into account your remark. We have added a comparative table with numerical data to the new version of the document.
  3. This is our mistake! We have corrected figure 7 in the new version of the document.
  4. The novelty of the work lies in the use of the well-known Panoptic Segmentation neural networks, trained on the Cityscapes dataset, to segment accident objects during the construction of a 3D accident scene using UAV, as well as in clustering the obtained 3D scene textures to detect objects on a 3D scene and the method of automatic distance determination. between objects, which consists in calculating the distance between clustered objects as a minimum of Euclidean distances between all points of equally clustered objects.

We hope that we have fully answered all your questions and comments.

Thank you for your cooperation!

Reviewer 3 Report

The authors proposed an automated approach for reconstruction of the accident site using Gazebo simulator and deep segmentation networks. The approach is interesting. The literature is well reviewed and all the steps are well-explained using appropriate figures. However, i have some concerns regarding this work.

1- I believe the vehicle pose estimation is one of the most important task for accident site reconstruction. The authors should clarify why this is missing in their work.

I also have some minor concerns:

line 363: X should be in bold font

line 364,367: V should be in bold font

line 364: which columns are principal components ==> which columns are principal directions

Author Response

Thanks for your feedback and recommendations! They are very important to us, because in this way we can do our work much better. Let us answer your comments.

  1. Yes, you are right. We agree with you, and the vehicle pose estimation is one of the most important task for accident site reconstruction. However, the reconstructed crash scheme with recognized objects and marked distances between them should be sufficient for analysis by special services.
  2. We will definitely check the formulas and make adjustments based on your comment.

We hope that we have fully answered all your questions and comments.

Thank you for your cooperation!

Round 2

Reviewer 1 Report

Overall, the new revision improves the paper. Now the goal of the work is much easier to understand and the description is much better. However, the same problems with the essence of the work remains the same:

1) Author claims that the novelty lie in applying existing panoptic segmentation models (per frame) in order to segment the scene, and this should automate the process of scene analysis. But this an obvious continuation of previous work, without real scientific novelty in methods or approaches.

2) The work is validated on 20 synthetic scenes. They are quite different from real scenes. First of all, they are much more clear, without debris and other objects (like other cars), which are usually present on the scene. E.g. https://www.photomodeler.com/pm-applications/pub-safety-forensics/drone-ar-mapping/ How the methods will fair for real scenes we don't know. Also, how big will be the automation in this case is still unknown. Because what measurements should be made will be decided by the user. Also, the real 3D reconstruction of the scene will contain errors, which are not considered here.

3) Why 2D panoptic segmentation methods are used? If 3D reconstruction is available, then 3D segmentation methods are applicable, which has better accuracy for 3D scenes. Why they haven't been considered?

4) The considered panoptic segmentation methods are trained on Cityscapes dataset, with camera basically on top of the car. The UAV have significanlty different position relative to the scene, so the model is better to be trained on the corresponding data. 

Author Response

Thanks for your feedback and recommendations! They are very important to us, because in this way we can do our work much better. Let us answer your comments.

1) You are absolutely right, this work is a direct continuation of our work [Saveliev, A., Izhboldina, V., Letenkov, M., Аksamentov, E., & Vatamaniuk, I. (2020). Method for automated generation of road accident scene sketch based on data from mobile device camera. Transportation research procedia, 50, 608-613.]. In this paper, which is reviewed, we analyze the known segmentation approaches for further optimization and their use on the UAV. That is, in future work, we plan to exclude the stage of sending data to the server for data processing and process data directly on the UAV. In the context of this work, the novelty lies precisely in the synthesis of a unified approach based on widely known and well-established methods to solve particular subtasks, which are part of the complex task of reconstructing traffic accident scenes under the set conditions.

2) Yes, we agree with you. However, the initial stage of testing the proposed approach, namely, working with synthetic data, is important for testing the performance and effectiveness of the proposed solutions. Of course, a dataset with real aerial photographs should be collected, which is what we are doing at the moment. Collecting real data (we plan to collect more than 100 scenes) is a very time-consuming process, which depends on weather conditions, traffic situations, as well as bureaucratic issues of coordinating shooting with authorities. Shooting real scenes at real intersections and roads in the city must be carried out with extreme care in order to avoid blocking the traffic flow and the occurrence of a situation that entails traffic accidents.

3) Your remark is correct. However, as mentioned above, we plan to develop our research towards performing all processing on the UAV, the computing power of which may not be enough to process deep neural networks that perform segmentation of 3D objects. Nevertheless, as part of further work, it is also planned to test 3D scene segmentation models.

4) Your remark is correct, however, we have not found a publicly available data set containing an exhaustive selection of the classes of road objects of interest to us. At the moment, we are working on it: we are collecting a dataset from aerial photographs of more than 100 scenes of real accidents for further training of our system and testing the proposed method on real data.

We hope that we have fully answered all your questions and comments.

Thank you for your cooperation!

Reviewer 2 Report

Accept in present form.

Author Response

Thanks for your feedback! It is very important to us, because in this way we can do our work much better.